# Temporal Investigation of the Maternal Origins of Fetal Gut Microbiota

**DOI:** 10.3390/microorganisms12091865

**Published:** 2024-09-09

**Authors:** Corrie Miller, Kayti Luu, Brandi Mikami, Jonathan Riel, Yujia Qin, Vedbar Khadka, Men-Jean Lee

**Affiliations:** 1Department of Obstetrics, Gynecology and Women’s Health, Division of Maternal Fetal Medicine, John A. Burns School of Medicine, University of Hawaii at Manoa, Honolulu, HI 96813, USA; millercb@hawaii.edu (C.M.); katyiluu@hawaii.edu (K.L.); mikamib@hawaii.edu (B.M.); jriel@hawaii.edu (J.R.); 2Department of Quantitative Health Sciences, John A. Burns School of Medicine, University of Hawaii at Manoa, Honolulu, HI 96813, USA; yqin@hawaii.edu (Y.Q.); vedbar@hawaii.edu (V.K.)

**Keywords:** fetal microbiome, neonatal microbiome, pregnancy microbial composition

## Abstract

In utero colonization or deposition of beneficial microorganisms and their by-products likely occurs through various mechanisms, such as hematogenous spread or ascension from the reproductive tract. With high-throughput sequencing techniques, the identification of microbial components in first-pass neonatal meconium has been achieved. While these components are low-biomass and often not abundant enough to culture, the presence of microbial DNA signatures may promote fetal immune tolerance or epigenetic regulation prior to birth. The aim of this study was to investigate the maternal source of the neonatal first-pass meconium microbiome. Maternal vaginal and anal samples collected from twenty-one maternal–infant dyad pairs were compared via principal component analysis to first-pass neonatal meconium microbial compositions. Results demonstrated the greatest degree of similarity between the maternal gut microbiome in the second and third trimesters and vaginal microbiome samples across pregnancy, suggesting that the maternal gut microbiota may translocate to the fetal gut during pregnancy. This study sheds light on the origin and timing of the potential transfer of maternal microbial species to offspring in utero.

## 1. Introduction

The identification of bacteria through DNA signatures in first-pass meconium and amniotic fluid has challenged traditional assumptions that the in utero environment is sterile [1,2,3,4,5]. This has led to a rapid rise in research investigating the fetal and neonatal gut microbiome. The microbiome is understood to have a significant role in immune regulation and the development of disease, making the neonatal microbiome critical for early life [6,7,8]. Several studies have identified the initial neonatal gut microbial composition to be associated with a wide range of neonatal conditions including, but not limited to, jaundice [9,10], sepsis [11,12], impaired growth of preterm infants [13], and prematurity [14,15,16]. This signifies the importance of understanding the role of neonatal microbes in infant health, and their potential to impact future development.

Currently, the mechanism behind how the fetal gut microbiome is populated is unclear. Previously, only culturable organisms known to be pathogens had been identified within the uterine compartment; otherwise, the womb was thought to be ‘sterile’ [17]. With the advent of sequencing technologies, the presence of non-pathogenic, possibly beneficial, microbes have been detected in utero [18]. This suggests that the presence of microorganisms in the fetal gut is likely transmitted in the prenatal period [4]. The source of such microbes and their role is not well understood. Potential theories include organisms ascending from the cervicovaginal canal, or hematogenous translocation from the oral cavity and gastrointestinal system [19]. Many experts support the idea that this microbial presence is unlikely to be a functional living colonization, and that it is more likely that small messages are packaged and sent via various mechanisms, such as extracellular vesicles (ECVs), to aid in early immune life [20,21].

Researchers have examined the importance of the fetal gut microbiome in relation to in utero immune tolerance [18], immune system regulation [22], and methylation and epigenetic regulation of metabolic syndromes [23]. Early-life exposures to maternal microbiota may help prime the fetal gut for subsequent microbial colonization during delivery and breastfeeding. All of these stages are foundational to the establishment of a healthy infant and childhood microbiome, which has shown to be protective against the development of chronic diseases later in life, such as asthma, allergies, and obesity [24,25]. The potential to impact in utero microbial community establishment through environmental exposures and maternal diet further underscores the need to better characterize maternal and fetal microbiomes.

Overall, the maternal microbiome likely has a significant impact on how microbiota populate the fetal gut in utero. Studies on maternal diabetes [26,27] and maternal diet [28] suggest that the maternal gut microbiome has a significant impact on neonatal colonization and early-life metabolism. Other studies have demonstrated the neonatal meconium microbiome to more closely resemble placenta or amniotic fluid microbial signatures than those of the maternal vaginal and gut microbiome [29,30]. Thus, ongoing research is necessary to further understand the origin, significance, and function of neonatal microbes, as this may be a window to manipulating and intervening in fetal health prior to birth.

This study aimed to identify maternal covariates, including maternal microbial composition from the gut and vagina, that could predict first-pass neonatal meconium composition. We evaluated these exposures longitudinally across gestation to identify a possible point in gestation that the origins of the fetal gut microbiome are determined. With such information, more precise targeting of the maternal environment could benefit fetal and neonatal development.

## 2. Materials and Methods

Subjects were recruited from the Kapiolani Medical Center for Women and Children (KMCWC) in Honolulu, Hawaii, during scheduled clinic visits at the Fetal Diagnostic Center. Methods were previously described [31]. Inclusion criteria were: enrollment during the first trimester of pregnancy (<14 weeks 0 days gestation), age between 18 and 45 years old, primarily English speaking and English literate, and self-identification as Asian, non-Hispanic White, or Native Hawaiian on the intake registration information form. As race and ethnicity have been highly associated with microbial composition, the study design further targeted the four most common races/ethnicities in our community for enrollment: Japanese, non-Hispanic White, Filipino, and Native Hawaiian. Individuals who identified as multiracial or of other races/ethnicities were excluded to eliminate this confounding variable, and to best sample the most representative population. Pregnant women who planned to move away from Hawaii prior to delivery or to deliver at another hospital or other study institution, had multiple gestations, were currently incarcerated, or had pre-existing uncontrolled medical conditions (diabetes, hypertension, heart disease, chronic renal disease, systemic lupus erythematosus, hypothyroidism, history of bariatric surgery, history of an eating disorder, or inflammatory bowel disease) were also excluded. All subjects signed the Western Institutional Review Board approved written informed consent form in compliance with the Hawaii Pacific Health protocol (Protocol#2018-039). Voluntary written consent was obtained from all subjects involved in the study prior to the initiation of any study-specific procedures. Patient demographic data collected from electronic medical records included patient age, maternal race/ethnicity, presence of obesity (classified as Body Mass Index (BMI) > 30 mg/kg^2^), gestational weight gain (characterized as “in excess” or not, according to the Institute of Medicine guidelines relating to BMI [32]), and delivery mode (vaginal or cesarean delivery). Neonatal birth weight, fetal sex, pregnancy complications, and neonatal complications were also noted.

### 2.1. Participant Sampling

Participants filled out demographic survey data at study intake and provided self-collected gastrointestinal (GIT) and vaginal samples once during each trimester using Copan eSwabs with amine preservatives (Copan Diagnostics, Murrieta, CA, USA). The first swab collections were completed at the time of enrollment, at which point patients were at between 11 and 13 weeks gestation, in the office at the patient’s ultrasound appointment. Vaginal swabs were collected from the mid vagina after rotating for 15 s. Anal swabs were collected from within 2 cm of the anal opening, again after rotating swabs within the lower portion of the anus for 15 s. Second trimester swab collections were performed at 18–20 weeks gestation, again after an ultrasound appointment in the office. The third trimester swab collections were performed at 34 to 36 weeks gestation at home and returned via mail within 48 h. Neonatal meconium samples were collected within 24 h of delivery from the infant diaper using Copan eSwab, with several negative control swabs opened and exposed to the air within the specimen collection room. Specimens were frozen at −80 °C until they were ready to be processed.

### 2.2. Microbiome Sequencing

An AllPrep DNA/RNA Extraction Kit (Qiagen, Hilden, Germany) was used to isolate whole genomic DNA. Extractions were performed at the Institute for Biogenesis Research at the University of Hawaii at Mānoa. DNA isolates were then sent to the Epigenomic Core at John A. Burns School of Medicine for sequencing using ThermoFisher Scientific 16S rRNA primers (Ion Torrent 16S Metagenomics Kit; Thermo Fisher Scientific, Warrington, UK). V2–4–8, V3–6, and V7–9 primers were used for amplification of the hypervariable regions of the 16S rRNA gene from bacteria, which is able to provide species-level resolution [33].

Sequencing was performed using the Ion Genestudio S5 Sequencer (ThermoFisher Scientific) according to the manufacturer’s instructions. One “air swab” and one negative control, using sterile water, were prepared alongside the biologic samples in each batch. Maternal samples were run as one batch, and neonatal meconium samples were run separately. The 16S Metagenomics Kit analysis was conducted using Ion Reporter™ Software v5.18.4.0 (Thermo Fisher Scientific). Chimeric sequences were automatically identified and removed, and reads were mapped to the Greengenes v13.5 and MicroSEQ ID v3.0 reference databases. The Ion Torrent data analysis platform was then used to align sequence fragments and group operational taxonomic units (OTUs) at the family, genus, and species levels. Raw abundance values were subsampled at 10,000 reads per sample to control for variations in read numbers across samples, with subsampling performed on the species-level operational taxonomic unit (OTU) table. Samples with fewer than 10,000 total reads were excluded from the dataset. 

### 2.3. Data and Bioinformatic Analysis

The cohort was recruited as a pilot study with a sample size limited to 40 participants. This analysis included maternal samples from 21 paired infant stool samples. The microbial alpha diversity of maternal samples for each trimester was calculated using the Chao1, Shannon, and Simpson indices. To ensure robustness, α-diversity indexes following rarefaction were computed using the average of 10 rarefied values at a sequence depth of 15,927 reads. Beta diversity profiles were evaluated through principal component analysis (PCA) based on the Bray–Curtis distance matrix. Analysis was conducted separately for each ethnic group and trimester. 

In our analysis, we focused on the first two principal components that explained the majority of the variance in the data. These components were used to visualize the data in lower dimensional space, allowing us to identify patterns, clusters, and potential outliers among the samples. The proportion of variance explained by each principal component was used to assess the importance of each component in capturing the overall variability of the dataset. All PCA calculations and visualizations were performed using R Studio version 4.2.0. 

Beta diversity profiles were examined based on maternal characteristics, including GIT versus vaginal delivery, delivery mode (vaginal versus cesarean section), trimester of maternal sample collection, maternal obesity, and ethnicity. The correlation between maternal and neonatal alpha diversity indices was compared using Pearson correlation.

## 3. Results

From the original cohort of 41 participants, 21 maternal–infant dyad matched samples were obtained (see Appendix A (CONSORT Diagram)). Maternal GIT and vaginal microbiome samples from each trimester were compared to the corresponding neonatal meconium sample. Maternal and offspring demographics are shown in Table 1.

### 3.1. Sequencing Depth and Quality Control

Following quality filtering, five samples were excluded from the analysis due to having fewer than 1000 reads. For the remaining samples, a total of 28 million high-quality reads were retained across all sample types, with an average of 110,212 reads per sample (vaginal: 118,898, GIT: 70,470, and meconium: 250,354). “Air swab” and kit controls were found to have negligible sequence reads, and thus eliminated from the subsequent analysis. Rarefaction analysis confirmed that the sequencing depth was sufficient to capture the majority of microbial diversity in each sample, ensuring reliable downstream analysis.

### 3.2. Operational Taxonomic Unit (OTU) Clustering and Taxonomic Assignment

High-quality sequences were clustered into operational taxonomic units (OTUs) at 97% similarity using QIIME 2. Representative sequences for each OTU were aligned against the Greengenes database (v13.5) for taxonomic classification. A total of 950 OTUs were identified, spanning 13 phyla, 217 genera, and 577 species.

### 3.3. Microbial Diversity and Composition

#### 3.3.1. Beta Diversity by Principal Component Analysis

Neonatal meconium microbial composition was found to be more similar to maternal GIT versus vaginal microbiome by evaluation using principal component analysis (Figure 1). In Figure 1, the meconium microbiota, represented by red, are in closer proximity to maternal GIT samples than vaginal samples. This is also demonstrated among individual maternal–infant dyad pairs in Figure 2. Bray–Curtis distance matrix-derived PCA analysis was performed to evaluate the influence of maternal demographics, specifically maternal race and ethnicity and obesity status, on neonatal microbial composition (Figure 3). Race and ethnicity grouped more closely than obesity, as seen with the more divergent centroids between non-Hispanic White participants and Japanese participants. No differences were seen with regards to delivery, likely due to the small number of cesarean deliveries (four in total), all of which were performed during labor.

#### 3.3.2. Alpha Diversity

To further examine the similarity between the maternal GIT microbiome and neonatal meconium microbiome, the correlation of alpha diversity indices (Chao, Shannon, and Simpson) was compared (Figure 4). While there was a trend in Chao index diversity between mothers and infants, this was not statistically significant. 

### 3.4. Relative Abundance across Gestation

The relative abundances at the phylum and genus levels according to the sample site are demonstrated in Figure 5. Maternal GIT samples are shown on the left, with vaginal samples at each trimester in the middle of the figure, and meconium composition on the right. At the genera level, the top 10 most abundant species were noted at each sample site (GIT, vaginal, or meconium). Generally, the microbiota within the meconium most resembled the maternal GIT composition, as opposed to the maternal vaginal composition. There was an overlap of the most abundant species between all three sites; however, five genera were seen primarily only in the meconium: *Citrobacter*, *Eubacterium*, *Lactococcus*, *Proprionibacterium*, and *Pelmonas.*

## 4. Discussion

This study aimed to understand the timing at which bacterial DNA transfer occurs during gestation and from which maternal body site. Our findings suggest that the fetal microbiota, as represented by the first-pass neonatal meconium, is most similar to the gut microbiome of an infant’s mother, as opposed to her vaginal microbiome, and that the organisms are most similar to the mother’s gut microbiome in the second and third trimesters. This is demonstrated in the similarity as measured via beta diversity and OTU abundance. The microbial similarities observed between our 21 dyad maternal–neonatal pairs further add to the growing wealth of evidence in the literature supporting the existence of an in utero microbiome, with the maternal environment impacting the fetal and infant microbiome.

Our study is unique in its comparison of maternal body site samples across gestation. Other studies have looked specifically at maternal sources of in utero ‘colonization’ or bacterial DNA translocation to the fetus. He et al. and Liu et al. both compared maternal vaginal, gastrointestinal, placental membrane, and amniotic fluid samples [30,34]. Using PCA analysis, they found that the microbiota of the neonatal meconium was the most similar to the microbial DNA signatures found in the placenta and amniotic fluid. Liu’s study looked at the vaginal and cesarean births of 78 dyad pairs and found that meconium bacterial communities were most similar to the placenta and fetal membranes, independent of delivery mode, alluding to pre-delivery transfer of microbiota. Both studies collected samples immediately before or during delivery, and not across gestation. Another study also compared maternal body site microbial composition (vaginal, stool, and saliva 1–2 days before delivery) and the impact of GDM on the neonatal microbiome [35]. They noticed increased dysbiosis across maternal biomes and neonatal biomes in those with GDM, but did not compare the similarities between possible origins of neonatal meconium and maternal biomes.

Overall, this study is impactful because a better understanding of the in utero process of microbial transfer to a fetus could aid in improving microbial composition and possibly preventing poor neonatal outcomes at birth. Several studies have shown that neonatal outcomes may be predicted by meconium microbial composition. Dornelles et al. identified *Proteobacteria* as being more prevalent in the presence of clinical early-onset neonatal sepsis in preterm infants by looking at the first-pass meconium, suggesting that the first meconium microbiota is different in preterm neonates with and without clinical early-onset neonatal sepsis [12]. In a prospective study with 63 preterm infants born at less than 33 weeks gestational age, Terrazzan et al. found that meconium microbiome abundance was related to appropriate weight for gestational age and that meconium microbiomes differed between children who achieved head circumference catch-up by the sixth month of corrected age [13]. Korpela et al. associated the first-pass meconium organisms of 212 consecutive newborn infants with the presence of infantile colic, noting a decreased abundance of *Lactobacillus* [36]. Therefore, future research should focus on the maternal GIT microbiome as a means of predicting neonatal outcomes, and finding methodologies to improve them.

The authors acknowledge the limitations of this study. First, the study methods cannot directly link maternal and fetal (as represented by first-pass neonatal meconium) bacterial DNA signatures, as 16S rRNA sequencing is primarily used for taxonomic classification. Deeper sequencing with strain-level resolution could help link specific strains in a mother’s samples and the infant microbiome. Our study also did not include the analysis of other sites such as fetal membranes, the placenta, or the maternal oral microbiome, nor did it involve sampling maternal sites at the time of delivery. We also did not collect information on maternal lifestyle, hygiene, or other habits that may have impacted exposures. It also has to be taken into consideration that the types of microorganisms that “colonize” or inhabit any intestinal mucosa niche are somewhat different from the vaginal mucosa niche in adult pregnant women, and this likely contributed to the observed similarities in microbial communities. Nonetheless, the observed similarity between the microbiomes of the second and third trimesters strongly suggests a hematogenous spread rather than an ascension pathway. Other limitations include that the 16S rRNA sequencing technique restricts the understanding of the mechanistic profiles or functional genomics of microbes present. Future studies utilizing shotgun sequencing and strain-level identification will be critical in tracing bacterial origins between mother and infant [37], thereby deepening our understanding of this maternal–infant microbial transfer. Additional research looking into mechanisms of bacterial transfer via ECVs or other antigen-presenting pathways over time should be explored.

## 5. Conclusions

This work represents the first study to look at changes to the neonatal profile throughout trimesters. There is evidence that bacterial exposure into the fetal compartment occurs throughout pregnancy [38,39], and the mechanism by which this occurs is poorly understood. Our results support that the intrauterine environment is populated by hematogenous spread, most likely from the gastrointestinal tract, as phyla and genera were most similar between these two communities.

## Figures and Tables

**Figure 1 microorganisms-12-01865-f001:**
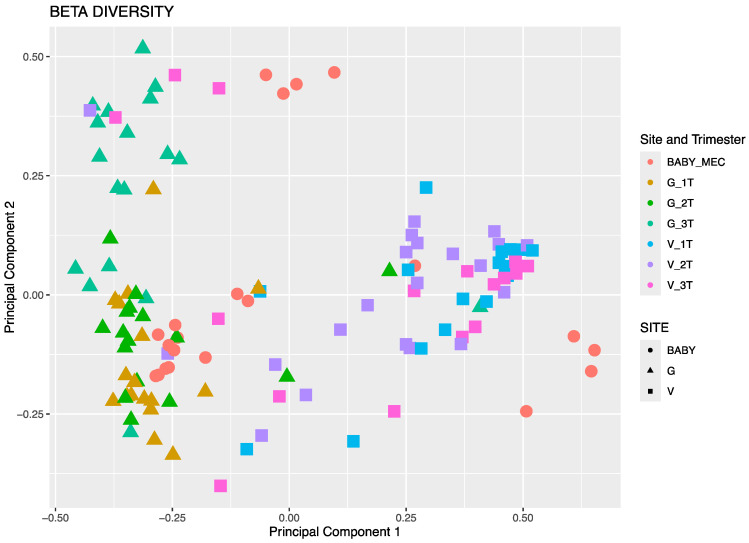
Principal component analysis derived from the Bray–Curtis distance matrix. Maternal gastrointestinal samples are depicted in mustard (first trimester), green (second trimester), and teal (third trimester), and maternal vaginal samples are depicted in blue (first trimester), purple (second trimester), and pink (third trimester). Neonatal meconium is depicted in red.

**Figure 2 microorganisms-12-01865-f002:**
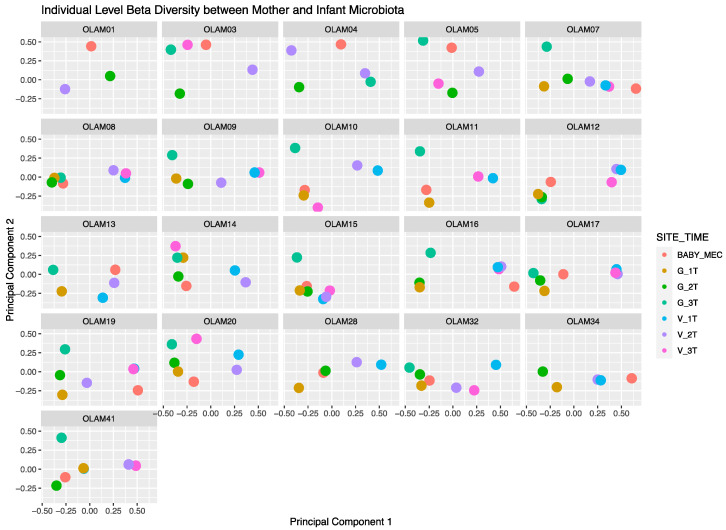
Principal component analysis derived from Bray–Curtis distance matrix for each maternal–infant dyad. Each facet represents a maternal–infant pair, with the Participant ID labeled at the top. Maternal gastrointestinal samples are depicted in mustard (first trimester), green (second trimester), and teal (third trimester), and maternal vaginal samples are depicted in blue (first trimester), purple (second trimester), and pink (third trimester). Neonatal meconium is depicted in red.

**Figure 3 microorganisms-12-01865-f003:**
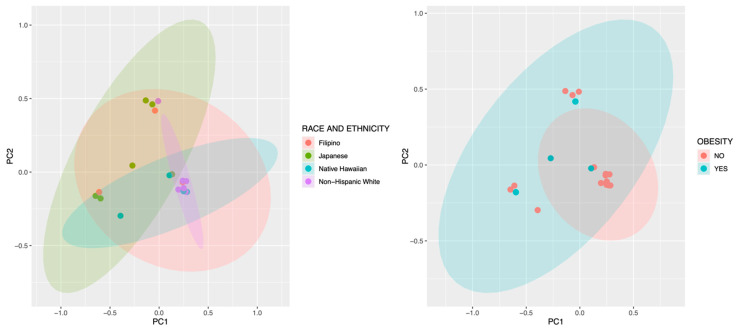
Beta diversity of neonatal meconium samples according to race and ethnicity, and maternal obesity status (BMI > 30 kg/m^2^).

**Figure 4 microorganisms-12-01865-f004:**
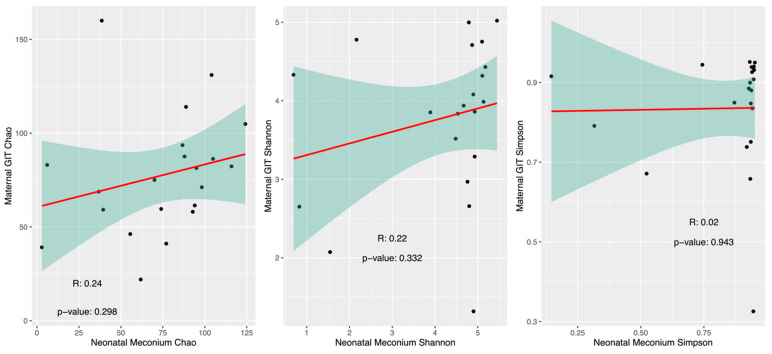
Pearson correlation of alpha diversity of maternal gastrointestinal (GIT) microbiome and neonatal meconium GIT microbiome using Chao, Shannon, and Simpson indices.

**Figure 5 microorganisms-12-01865-f005:**
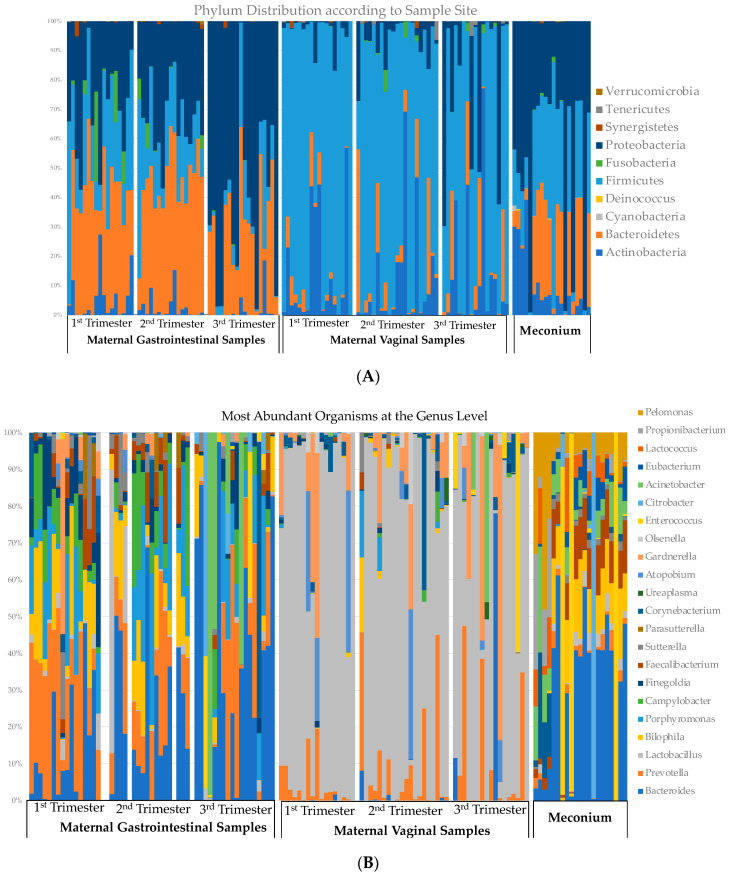
Variation at the phylum level (**A**) and genus level (**B**) of neonatal first-pass meconium and maternal gastrointestinal and vaginal samples according to trimester. The top 10 genera were picked for each sampled site, with much overlap. Panel B demonstrates organisms belonging to the top 22 genera.

**Table 1 microorganisms-12-01865-t001:** Maternal demographic characteristics. Maternal obesity defined as Body Mass Index (BMI, m/kg^2^) > 30; excess gestational weight gain defined as those participants who gained more than the recommended amount by the Institute of Medicine [32] according to BMI.

Baseline Demographics of Participants with Matched Neonatal Meconium Samples *n* = 21	
Maternal Age (years), Median [SD]	28.6 [5.4]
Maternal Ethnicity	
Filipino	5
Japanese	6
Native Hawaiian	4
Non-Hispanic White	6
Parity	
Nulliparous	8
Primiparous	10
Multiparous	3
Mode of Delivery	
Vaginal	17
Cesarean Delivery	4
Maternal Obesity	4/21 (19%)
Excess Gestational Weight Gain	5/21 (23.8%)
Pregnancy Complications	
Gestational Diabetes	2
Pregnancy Associated Hypertension	4
Preterm Birth	1
Neonatal Birth Weight (grams), Mean [SD]	3203.62 [561.7]

## Data Availability

Following the acceptance of our manuscript for publication, these datasets will be deposited into appropriate databases, including the NCBI Gene Expression Omnibus (GEO) database (https://www.ncbi.nlm.nih.gov/geo/), the NCBI Short Read Archives (SRA) (https://www.ncbi.nlm.nih.gov/sra), and MicrobiomeDB (https://microbiomedb.org/mbio/app) (accessed on 15 August 2024).

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
