# Peer review of "Temporal Investigation of the Maternal Origins of Fetal Gut Microbiota"

_microorganisms, 2024, doi:10.3390/microorganisms12091865_

Round 1
Reviewer 1 Report
Comments and Suggestions for Authors
Title.
It would be clearer to represent the purpose of the study. Its missing the relationship of the gestational to the fetal microbiome.
Abstract.
This sentence is unclear. “Results demonstrate greatest similarity between maternal gut microbiome in the 2nd and 3rd trimesters, compared to vaginal microbiome samples across pregnancy.” It does not seem to flow from your purpose which is to compare maternal body site microbial composition to neonatal first pass microbiome.
Introduction.
Line 50. Word choice problems- Currently, the origin of the fetal microbiome is unclear.
Line 53. Awkward sentence. Possible edits….suggest that the maternal microbiome has a significant impact…
Line 61. Personification. Studies have theorized…
Lines 63-65 Awkward sentence. Maternal to fetal microbiome is unclear. What you are referring to is the development of infant microbiome which is acquired from the mother during pregnancy, labor and birth and breastfeeding.
Lines 65-67. Unclear. How can you say you can control exposures in humans? It sounds like you are referring to mice in cages.
Lines 69-74 very clear purpose!
Materials and Methods.
Line 76. Subjects were recruited from….
Line 78. Included is redundant of inclusion criteria.
Lines 78-83. Please clarify. Did you exclude other races or are you describing who you enrolled?
*Did you collect diet and lifestyle (e.g. sexual practices, vaginal cleansing) information or other covariates of vaginal and GI microbiome?
Results
*A participant flow diagram would help the reader understand how you recruited and retained participants.
*Nulliparous (never gave birth) and primiparous (gave birth once) are generally considered the same thing since it is considered while the individual is in the current pregnancy. Is there some reason you differentiated between these two?
*This paragraph is unclear. It may be a word choice problem. “Neonatal meconium microbial composition was found to be more similar to maternal GIT compared to vaginal microbiome by evaluation of principal component analysis (Figure 1) and Pearson correlation (Figure 2).” It would be clearer if you used the word “verses”.
* While your sentence in the findings does explain it, Figure 1 on its own is difficult to interpret. The legends at the bottom and side of the figure have no intuitive meaning to the reader as they refer to laboratory testing.
*Figure 3. Figure legends on side and bottom of graph are not interpretable. It would be clearer to just use simple participant numbers for each block instead of the complex letter/number for each.
*Figure 4. The long participant IDs are impossible to read.
*Figure 5. What are PC1 and PC2?
Discussion
*This section needs better organization and refinement.
*The first paragraph belongs in the intro/lit review as it is not a discussion of your findings.
*It would be clearer to begin with a summary of major study findings and in your case to indicate that you did not link maternal microbiome to neonatal first pass mec (as you stated later in lines 265-267.
*Lines 252-263. Most of this paragraph of discussion goes beyond the findings. This is a small observational pilot study.
*Please add implications for future research.
*Line 283 awkward sentence/word choice problems, “in the process of being understood”…
Comments on the Quality of English Language
Multiple problems with word choice, sentence structure and missing words.
Author Response
Title.
It would be clearer to represent the purpose of the study. Its missing the relationship of the gestational to the fetal microbiome.
Thank you for this comment. This is very imperative component of the manuscript and we welcome the feedback. We have modified the title as such:
“Temporal Investigation of the Maternal Origins of Fetal Microbial Composition.”
Abstract.
This sentence is unclear. “Results demonstrate greatest similarity between maternal gut microbiome in the 2nd and 3rd trimesters, compared to vaginal microbiome samples across pregnancy.” It does not seem to flow from your purpose which is to compare maternal body site microbial composition to neonatal first pass microbiome.
Thank you for your comment and perspective. From your statement, it is clear we did not define the objective of the study well – as also mentioned in the title. We have changed line 29 to state that the overall objective of the study is to understand which maternal body site is the source of neonatal first pass meconium microbiome. We have also clarified line 34 regarding what the results demonstrate: - that the possible source of fetal gut microbiome ( as represented by first-pass meconium) is the maternal gut.
Introduction.
Line 50. Word choice problems- Currently, the origin of the fetal microbiome is unclear.
We have rephrased this sentence and the subsequent concepts to better clarify. We have also incorporated the paragraph from the discussion that was recommended to be moved to introduction here.
Line 53. Awkward sentence. Possible edits….suggest that the maternal microbiome has a significant impact…
Thank you – this sentence has been reworded
Line 61. Personification. Studies have theorized…
Thank you – this sentence has been reworded to state that “Researchers have examined”
Lines 63-65 Awkward sentence. Maternal to fetal microbiome is unclear. What you are referring to is the development of infant microbiome which is acquired from the mother during pregnancy, labor and birth and breastfeeding.
Thank you for pointing this out. We wanted to relay the concept that the fetal microbiome may lay the groundwork or prime the fetal gut to establish the microbiome during post-natal life, and have reworded and expounded on this concept.
Lines 65-67. Unclear. How can you say you can control exposures in humans? It sounds like you are referring to mice in cages.
We have changed this word “control” to “impact”
Lines 69-74 very clear purpose!
Materials and Methods.
Line 76. Subjects were recruited from….
“To” has been changed to “from”
Line 78. Included is redundant of inclusion criteria.
Included was omitted.
Lines 78-83. Please clarify. Did you exclude other races or are you describing who you enrolled?
Thank you – this area has been clarified to justify exclusion of other races due to the strong association of microbial communities with race/ethnicity.
*Did you collect diet and lifestyle (e.g. sexual practices, vaginal cleansing) information or other covariates of vaginal and GI microbiome?
We did not collect additional lifestyle behaviors, and have listed this in the limitations of the study.
Results
*A participant flow diagram would help the reader understand how you recruited and retained participants.
Thank you for this suggestion, we have added a CONSORT diagram to the supplementary material.
*Nulliparous (never gave birth) and primiparous (gave birth once) are generally considered the same thing since it is considered while the individual is in the current pregnancy. Is there some reason you differentiated between these two?
We have identified differences in previous research that parity, even if one previous birth, had variation on cytokine production in mothers, highlighting the immune shifts that may occur after giving birth once. Other studies have noted the differences in the gut microbiome related to parity [1]. The delineation between 1 birth versus many births may be more questionable, but we thought it important to show the distribution of parity.
*This paragraph is unclear. It may be a word choice problem. “Neonatal meconium microbial composition was found to be more similar to maternal GIT compared to vaginal microbiome by evaluation of principal component analysis (Figure 1) and Pearson correlation (Figure 2).” It would be clearer if you used the word “verses”.
Thank you for the suggestion, we have changed the word “compared” to “versus”
* While your sentence in the findings does explain it, Figure 1 on its own is difficult to interpret. The legends at the bottom and side of the figure have no intuitive meaning to the reader as they refer to laboratory testing.
Thank you, we have further explained the method of principal component analysis in the methodology section, relabeled the X and Y axes for this figure, and further clarified in the caption of the figure that the X and Y axes represent 1st and 2nd principal components.
*Figure 3. Figure legends on side and bottom of graph are not interpretable. It would be clearer to just use simple participant numbers for each block instead of the complex letter/number for each.
Thank you – the labels for X and Y axes have been changes to Principal Component 1 and 2, and the caption has been modified to explain facet labels.
*Figure 4. The long participant IDs are impossible to read.
Thank you, per suggestion of the other reviewer, this graph was omitted.
*Figure 5. What are PC1 and PC2?
We have clarified the methods of principal component analysis, and clarified in within figure 5 caption PC1 and PC2 = principal component 1 and 2.
Discussion
*This section needs better organization and refinement.
Thank you for this perspective, we have significantly re-organized the discussion as suggested.
*The first paragraph belongs in the intro/lit review as it is not a discussion of your findings.
These concepts have been incorporated into the introduction.
*It would be clearer to begin with a summary of major study findings and in your case to indicate that you did not link maternal microbiome to neonatal first pass mec (as you stated later in lines 265-267.
Thank you for this comment. We have restructured the discussion as such, and better clarified that statement – that the methodology used cannot directly link, with absolute certainty, the definitive source of fetal microbiota, but this study suggests the most likely source being through hematogenous spread instead of ascending vaginal spread.
*Lines 252-263. Most of this paragraph of discussion goes beyond the findings. This is a small observational pilot study.
Thank you, we have tried to make the discussion more relevant to the confines of the study design.
*Please add implications for future research.
We have included suggestions for future research at the end of the last discussion paragraphs, lines 292-294, and 209-315
*Line 283 awkward sentence/word choice problems, “in the process of being understood”…
Thank you – we have reworded to state, “and the mechanism by which this occurs is poorly understood”
Comments on the Quality of English Language: Multiple problems with word choice, sentence structure and missing words.
We have had multiple reviewers re-read the revised copy of the manuscript.
- Berry, A.S.F., et al., Remodeling of the maternal gut microbiome during pregnancy is shaped by parity. Microbiome, 2021. 9(1): p. 146.

Reviewer 2 Report
Comments and Suggestions for Authors
This isn't the first study to correlate microbiota in neonatal first-past meconium samples and maternal gut and vaginal samples. The significance of this study is to further addressing which tri-mester maternal samples would affect the formation of microbiota in neonatal guts.
The method of 'Data and Bioinformatic Analysis' isn't clear. What tools and parameters you used to filter and process generated sequences and then characterise the microbiota? Did you include a negative control in every batch of extraction and sequencing?
The results of sequencing depth isn't reported. Did you compute a rarefaction curve to check if the sequencing depth reach sampling sufficient?
I'd suggest to remove Table 2 and make Figure 4 more informative by creating two bar charts of relative abundance of top bacterial communities at phylum and genus levels. 16S sequencing isn't good enough for species level characterisation.
The only conclusion you can make is that the microbiota in maternal guts showed significant higher correlations than those in vaginal samples with the neonatal first-past meconium microbiota.
Author Response
This isn't the first study to correlate microbiota in neonatal first-past meconium samples and maternal gut and vaginal samples. The significance of this study is to further addressing which trimester maternal samples would affect the formation of microbiota in neonatal guts.
The method of 'Data and Bioinformatic Analysis' isn't clear. What tools and parameters you used to filter and process generated sequences and then characterise the microbiota? Did you include a negative control in every batch of extraction and sequencing?
Thank you, we acknowledge that we should further clarify and provide details about the description of the methodology and have expanded the Ion Torrent platform with language in the methods section (Lines132-152
The results of sequencing depth isn't reported. Did you compute a rarefaction curve to check if the sequencing depth reach sampling sufficient?
Yes, thank you for this comment, we have added additional details about sequencing depth and average read counts for the various samples. The rarefaction curve was checked at sequence number 15,927 (lines 158-160), and found to be sufficient (lines 188-190)
I'd suggest to remove Table 2 and make Figure 4 more informative by creating two bar charts of relative abundance of top bacterial communities at phylum and genus levels. 16S sequencing isn't good enough for species level characterisation.
Thank you for this suggestion. Understanding that 16s with typical coverage using V3 or V4 primers is typically insufficient for speciation, the methodology used with spanning V2-V9 has had good results as speciation. Nonetheless, we understand the concerns from the reviewer, and have modified the graphs as such.
Comparing the Phylum and genus of top bacterial abundance in each sample is an excellent suggestion. Thank you for this recommendation. We have modified Figure 4 as such.
The only conclusion you can make is that the microbiota in maternal guts showed significant higher correlations than those in vaginal samples with the neonatal first-past meconium microbiota.
Round 2
Reviewer 1 Report
Comments and Suggestions for Authors
Excellent revisions. Readability and relevance have been significantly improved.
Author Response
Thank you for your comments and perspective to help significantly improve the manuscript.